# Intermittent Fasting Promotes Weight Loss without Decreasing Performance in Taekwondo

**DOI:** 10.3390/nu15143131

**Published:** 2023-07-13

**Authors:** Ronaldo Angelo Dias da Silva, Leszek Antoni Szmuchrowski, João Paulo Pereira Rosa, Marcos Antônio Pereira dos Santos, Marco Túlio de Mello, Lucas Savoi, Yves Ferreira Porto, Francisco de Assis Dias Martins Júnior, Marcos Daniel Motta Drummond

**Affiliations:** 1Laboratório de Nutrição e Treinamento Esportivo, Universidade Federal de Minas Gerais, Belo Horizonte 31270-901, MG, Brazil; ronaldoangelo@ufmg.br (R.A.D.d.S.); lucassavoi@hotmail.com (L.S.); yves.porto@yahoo.com.br (Y.F.P.); francisco.junior@aluno.ufop.edu.br (F.d.A.D.M.J.); 2Laboratório de Avaliação da Carga, Universidade Federal de Minas Gerais, Belo Horizonte 31270-901, MG, Brazil; leszek_br@yahoo.com.br; 3Department of Physical Education, Institute of Biosciences, São Paulo State University (UNESP), Rio Claro 13506-900, SP, Brazil; jpseipai@hotmail.com; 4Nucleus of Study in Physiology Applied to Performance and Health, Department of Biophysics and Physiology, Federal University of Piauí, Teresina 64049-550, PI, Brazil; marcosedfisio@gmail.com; 5Centro de Estudos em Psicobiologia e Exercício, Universidade Federal de Minas Gerais, Belo Horizonte 31270-901, MG, Brazil; tmello@demello.net.br

**Keywords:** sports nutrition science, martial arts, weight loss, sports combat, elite athletes

## Abstract

Intermittent fasting (IF) is commonly used by combat sports athletes for weight loss. However, IF can decrease performance. This study aimed to investigate the effect of IF on total body mass (TBM) and Taekwondo performance. Nine athletes (seven male, two female; 18.4 ± 3.3 years) underwent 4 weeks of 12 h IF. TBM, countermovement jump (CMJ), mean kicks (MK), and total number of kicks (TNK) were compared weekly. Performance was measured in the fed state (FED) and fast state (FAST). Results showed decreased TBM in week 1 (62.20 ± 6.56 kg; *p* = 0.001) and week 2 (62.38 ± 6.83 kg; *p* = 0.022) compared to pre-intervention (63.58 ± 6.57 kg), stabilizing in week 3 (62.42 ± 6.12 kg), and no significant change in week 4 (63.36 ± 6.20 kg). CMJ performance in week 1 was lower in FED (35.26 ± 7.15 cm) than FAST (37.36 ± 6.77 cm; *p* = 0.003), but in week 3, FED (38.24 ± 6.45 cm) was higher than FAST (35.96 ± 5.05 cm; *p* = 0.047). No significant differences were found in MK and TNK in FSKTmult. RPE, KDI, and HR were similar between FED and FAST (*p* < 0.05). [LAC] was higher post-test compared to pre-test (*p* = 0.001), with higher concentrations in FED than FAST (*p* = 0.020). BG was higher in FED than FAST (*p* < 0.05) before physical tests. Therefore, IF promotes decreased TBM without decreasing performance.

## 1. Introduction

In combat sports, there are numerous strategies for rapid reduction of total body mass (TBM), such as water restriction, use of laxatives, immersion in hot water baths, sauna sessions, increased quantity and duration of training sessions, and performing exercises with heavy clothing and/or plastic suits [1]. However, these strategies can lead to detrimental effects on athletic performance, including muscle mass reduction, decreased overall and specific endurance, impaired training adaptations, decreased concentration capacity, difficulty in decision-making, mood alterations such as increased irritability and demotivation, and increased discomfort and subjective perception of effort following training sessions [2]. Therefore, the adoption of nutritional strategies such as hypocaloric diets and intermittent fasting can be less harmful alternatives for reducing body mass in this population.

Intermittent fasting (IF) is the partial or total abstinence of food intake for a predetermined period of 12 to 48 h with little or no caloric intake interspersed with non-restricted feeding [3]. Therefore, it is common for Taekwondo athletes to use it as a nutritional strategy for weight loss [4,5]. In eutrophic individuals, this strategy can decrease the TBM by 2% within a few weeks [6]. Such a lower TBM may result from behavioral and physiological changes that regulate appetite and result in lower caloric intake in the hours of fasting [7] that is not compensated in the non-restricted feeding period [8].

However, fasting exerts some effects that might lower sports performance, such as sleep disorders, hypohydration, low motivation, and reduced muscle glycogen stores [9,10]. The caloric restriction that is common in fasting can lower the mitochondrial intrinsic activity without altering the mitochondrial content in skeletal human muscle [11]. Moreover, longer IF periods can increase cortisol levels and stimulate hunger, which, in turn, can cause the TBM to be either maintained or increased [12].

Despite the various IF protocols [3], several studies have reported solely the effects of Ramadan fasting in combat sports [10,13,14,15,16,17,18]. Some studies have found no effect of Ramadan fasting on performance in combat sports and a non-significant decrease in TBM measured before, during, and at the end of Ramadan [10,14]. However, in others, Ramadan fasting decreased performance in combat sports [13,15,16,17,18]. Such a finding was reported especially upon the application of tests in the afternoon, thus with accumulated fasting, followed by lower TBM and energy intake [13,15,16,17,18]. In Taekwondo, Memari et al. [17] found that Ramadan fasting not only decreased the TBM but also the performance in agility Taekwondo tests. Pak et al. [18] applied a specific taekwondo test during IF and reported a lower number of kicks at the end of Ramadan. Therefore, studies on the effect of IF on weight loss and sports performance show conflicting results. However, Ramadan does not represent the reality of Taekwondo athletes from several countries. In this regard, the IF protocol adopted by these athletes is different and requires specific investigations.

Silva et al. [7] investigated the acute effect of a 12 h fast with breakfast omission on weight loss and performance in taekwondo athletes. The results showed acute weight loss and non-significant effects on performance in countermovement jump (CMJ) height and frequency speed of kick test multiple series (FSKTmult). We could not find any studies that investigated IF in Taekwondo training other than those that concerned Ramadan. In this sense, considering that even though IF can reduce TBM in Taekwondo athletes, the literature reports are limited regarding its influence on training performance; therefore, the potential chronic effects of IF on Taekwondo should be further investigated.

Accordingly, given the results of previous studies, our hypothesis is that IF can promote weight loss without reducing performance in Taekwondo. Therefore, this study aimed to examine the effect of a 12 h IF for four weeks on weight loss and performance in Taekwondo.

## 2. Methods

### 2.1. Subjects

Nine Taekwondo athletes (seven male and two female individuals) (18.4 ± 3.3 years) participated in this study, with a mean of 9.2 ± 3.4 years of experience in the modality and participation in international championships. Our inclusion criteria covered those who were competing and kept their body masses stable for at least two months before the research period. In addition, the athletes should be aiming for a lower TBM to adjust to the weight category. Thus, our sample represents a specific category of Taekwondo athletes. Additionally, the athletes must have been undergoing an isocaloric diet before the intervention period, and no injuries in the lower limbs must have occurred over the past three months. Our exclusion criteria considered the volunteers whose TBM had no variation over the first two weeks of study and who suffered any joint and/or muscular injury in the lower limbs during the study period. In addition, all athletes who did not adhere to the 12 h fast over the four weeks of data collection were also excluded. This study was approved by the Ethics and Research Committee of the Federal University of Minas Gerais (protocol: 2.582.067).

### 2.2. Experimental Design

This study proposes an experimental paired trial covering a total of six weeks. Initially, the first two weeks covered the familiarization, reliability, and performance of the baseline (BL) measure. In the following four weeks, the athletes performed the same BL tests but in eight moments and two different situations: on four Monday afternoons, approximately two hours after lunch in the fed state (FED), and on four Saturdays before skipping breakfast in the fasted state (FAST). Figure 1 illustrates the study design.

The CMJ and the FSKTmult tests were performed. All athletes answered the rating of perceived exertion (RPE) at the end of the test sessions. The lactate concentration [LAC] was measured pre- and two minutes post-FSKTmult using an Accutrend Plus lactometer (Roche Diagnostics, Brazil Ltda., Sao Paulo, Brazil). To avoid performing the tests in a state of hypoglycemia and to verify whether the athletes were in the FED or FAST state, we measured the values of both pre- and post-test blood glucose [BG] concurrently with lactate [LAC] using a FreeStyle Optium Neo glucometer (Abbott Laboratórios do Brasil Ltda., Rio de Janeiro, Brazil).

The training frequency of the athletes reached six sessions per week throughout the six weeks. Five sessions were performed from Monday to Friday in the afternoon (2:00 p.m.), with a 24 h interval between sessions. The sixth training session took place on a Saturday morning (8:00 a.m.), with a 14 h interval from the Friday training session.

The IF protocol applied over the four-week intervention consisted of 12 h of time-restricted feeding, starting at night, with breakfast omission, interspersed with periods of non-restricted feeding. The volunteers were instructed to maintain their eating habits during the feeding period, and no nutrition intervention was performed. On days of testing in the fed state, the pre-workout meal was consumed between 1 to 2 h before the start of the tests.

During the six weeks, the athletes’ food intake was recorded weekly over three non-consecutive days [19], followed by a supplementary dietary recall [20], to measure the total calories ingested and the macronutrient composition. Prior to the beginning of the study, the volunteers received explanations and training to record their food intake using homemade measures and photographic records. The supplementary dietary recall was performed based on the weekly food intake record to identify possible errors in the process carried out by the volunteers, such as food and meal identifications, homemade measurements, and timing. Each week, the dietary profile of each volunteer was determined using the supplementary dietary recall, which was adopted to monitor and compare the nutritional intake between the weeks of the experimental intervention. The dietary profile was determined based on the daily total caloric intake and the intake of macronutrients: carbohydrates, lipids, and proteins. All values were normalized by TBM. These procedures were performed using the DietBox app (Dietbox Informática Ltda., version 7.8.1, Porto Alegre, Brazil) and the following tables, in order of preference: the Brazilian Table of Food Composition (TACO), the Nutritional Composition Table of Foods Consumed in Brazil (IBGE), and the Sonia Tucunduva Philippi Table.

### 2.3. Familiarization and Reliability

The athletes participated in two familiarization sessions in the first week and two reliability sessions in the second week, respecting a 48 h interval. In each session, we applied a sequence of two sets of eight CMJs with a one-minute interval between jumps and 10 min between the sets. If in a single session, the first and second sets presented a statistically significant difference, then an addition set with eight CMJs was performed until two consecutive sets showed no statistically significant difference [21]. Familiarization was considered successful when there was no statistically significant difference between the means of jump heights in two consecutive sessions [21]. All sessions were performed using PLA3–1D-7KN/JBA Zb (Staniak; Warsaw, Poland, 1000 Hz with 1 N accuracy).

The FSKTmult test was applied ten minutes after the CMJ test and encompassed three rounds of five sets with 10 s of stimuli, 10 s of interval rest between sets, and a one-minute interval between rounds [7]. The test was conducted using Boomboxe^®^ (Simulacare, São Paulo, Brazil), and the athletes were instructed to perform the maximum number of kicks possible. During the FSKTmult test, all athletes used a heart rate sensor (Polar H10, Polar Electro Brasil, Ltda., Sao Paulo, Brazil) to have both the mean (HRmed) and maximum (HRmax) heart rates measured, and at the end of the test, they answered the RPE immediately. The performance analysis involved registering the total number of kicks (TNK) and the mean of kicks performed (MK) during three rounds. Additionally, the kick decrease index (KDI) was calculated using the following equation proposed by Santos and Franchini [22].
(1)$$\mathrm{KDI}=\left[1-\frac{\mathrm{FSKT}1+\mathrm{FSKT}2+\mathrm{FSKT}3+\mathrm{FSKT}\;4+\mathrm{FSKT}5}{\mathrm{best}\;\mathrm{FSKT}\;\times \;\mathrm{number}\;\mathrm{of}\;\mathrm{sets}}\right]\times 100$$

### 2.4. Performance Protocols

The BL was performed 48 h after the last reliability session with the athletes in FED (Figure 1). The CMJ height in BL was assessed by applying a set of five jumps with a one-minute interval rest between them. Ten minutes later, the FSKTmult test was performed following the same familiarization procedures.

The test sessions in FED were conducted on Mondays, 48 h after the last training session, whereas the tests in FAST took place on Saturdays after the 14 h rest interval from the Friday training session (Figure 1). For the tests in FAST, the athletes were instructed to have their last meal of the day 12 h before. The same test sessions and procedures as the BL were applied.

### 2.5. Total Body Mass Measures

The TBM was measured in FAST (pre-intervention) and then weekly using a digital scale (Welmy^®^, Santa Bárbara d’Oeste, Brazil) on Saturdays and before the tests. To prevent potential small, systematic, and predictable errors due to changes in the balance of body fluids, the athletes were instructed not to ingest large amounts of water, to urinate at least 30 min before, and not to consume alcoholic beverages or perform any physical exercise within 24 h other than the Taekwondo training session.

### 2.6. Statistical Analysis

The normality and sphericity of the data were verified using the Shapiro–Wilk and Mauchly tests. If neither the normality nor the sphericity assumptions were met, non-parametric tests and the Greenhouse–Geisser correction were applied. The stabilization of both the CMJ and FSKTmult tests during the familiarization sessions was analyzed through the paired student *t*-test [21]. Reliability was verified by a two-way mixed intraclass correlation coefficient (ICC) [23]. A two-way ANOVA with repeated measures was applied for the TBM regarding the amount of calories ingested and the proportion of macronutrients throughout the weeks. This procedure (time x situation) was used to compare the performance variables (CMJ, TNK, MK, HRmed, HRmax, RPE, and KDI). [BG] and [LAC] in different situations were analyzed through a three-way ANOVA with repeated measures (week x situation x moment). Bonferroni’s post hoc test was used to identify where differences occurred. The Cohen’s d effect size for paired samples was verified based on the partial ETA (ŋp^2^) square. The effect size was considered small, medium, or large if d = 0.20 or ŋp^2^ = 0.04, d = 0.50 or ŋp^2^ = 0.25, and d = 0.8 or ŋp^2^ = 0.64, respectively [24,25]. All data are presented as mean, standard deviation, and 95% confidence interval (CI95%). All analyses considered a significance level of α = 0.05. All statistical analyses were performed on SPSS version 20.0.

## 3. Results

### 3.1. Familiarization and Reliability

The first (T(8) = 0.471, *p* = 0.689, d = 0.03: set 1 = 30.61 ± 3.65 cm, set 2 = 29.75 ± 7.13 cm) and second familiarization sessions (T(8) = 0.940, *p* = 0.378, d = 0.08: set 1 = 30.95 ± 8.32 cm, set 2 = 28.70 ± 5.66 cm) had similar CMJ height, with a small effect. Likewise, no difference was found in the CMJ height between the first and second sessions (T(8) = 0.214, *p* = 0.833, d = 0.01, session 1 = 30.18 ± 5.49 cm, session 2 = 29.82 ± 6.97 cm), and the effect size was also considered small. The FSKTmult test indicated no statistically significant difference for the total mean of kicks in the first familiarization session (92.44 ± 8.79) (T(8) = −0.595; *p* = 0.569; d = 0.02) regarding the second session (93.33 ± 6.87).

The reliability test for the CMJ found a significant and excellent correlation (ICC = 0.97; *p* = 0.001; CI95% = 0.90–0.99) between the jumps of the first (37.47 ± 5.93 cm) and second sessions (37.20 ± 6.48 cm). The FSKTmult indicated a significant correlation to the mean number of total kicks in the first session (89.00 ± 8.2), ranked as good (ICC = 0.89; *p* = 0.001; CI95% = 0.57–0.97) compared with the mean number of total kicks in the second session (91.11 ± 6.67).

### 3.2. Total Body Mass

The TBM decreased with a large effect size throughout the weeks (F(4) = 7.120; *p* = 0.006; ŋp^2^ = 0.47). The BonFerroni’s test detected reductions in the first with a small effect size (62.200 ± 6.567 kg; *p* = 0.001; 95%CI = 0.60, 2.16; Cohen’s d = −0.05) and second weeks (62.389 ± 6.833 kg; *p* = 0.022; 95%CI = 0.15, 2.23; Cohen’s d = −0.04) of fasting compared with the pre-intervention (63.584 ± 6.576 kg); however, it was later stabilized and kept unchanged in the third (62.422 ± 6.128 kg) and fourth weeks (63.369 ± 6.204 kg) (Figure 2).

### 3.3. CMJ

No difference was indicated regarding the CMJ either between the intervention weeks (F(3) = 1.232; *p* = 0.320; ŋp^2^ = 0.13) or between situations (F(2) = 0.994; *p* = 0.392; ŋp^2^ = 0.11). However, there was a difference in the week x situation interaction, with a large effect size (F(6) = 7.040; *p* = 0.001; ŋp^2^ = 0.46). The BonFerroni posthoc test detected a difference in the FED compared with the BL (*p* = 0.049; 95%CI = 0.01–3.33) concerning the FAST (*p* = 0.003; 95%CI = −3.33, −0.87) in the first week; however, no difference was found in the second week. In the third week, the BonFerroni posthoc test detected a difference between the FED and FAST (*p* = 0.047; 95%CI = 0.03, 4.52), whereas in the fourth week, a difference between the situations was found. Furthermore, the CMJ for the FED in the third week was higher than that in the first week (*p* = 0.032; 95%CI = −5.72, −0.24), which also occurred for the fourth week compared with the first week (*p* = 0.050; 95%CI = −5.37, 0.01). Table 1 shows the respective results.

### 3.4. FSKTmult

The results of the FSKTmult test indicated no difference in the MK or the TNK between the weeks (F(3) = 3.744; *p* = 0.074; ŋp^2^ = 0.31), between the situations (F(2) = 2.610; *p* = 0.134; ŋp^2^ = 0.24), or in the week x situation interaction (F(6) = 2.068; *p* = 0.140; ŋp^2^ = 0.20) (Table 2). Compared with the BL, the results of RPE showed significantly higher values (*p* = 0.020) in the FAST for the second week, the FAST and FED for the third week, and the FAST and FED for the fourth week. Additionally, the RPE in the fourth week was higher than the BL (*p* = 0.001), as well as in the second (*p* = 0.033) and third weeks (*p* = 0.010), with a large effect size (ŋp^2^ = 0.65). Regarding the mean KDI, no difference occurred between the weeks (F(3) = 1.366; *p* = 0.277; ŋp^2^ =0.14), between situations (F(2) = 0.036; *p* = 0.965; ŋp^2^ = 0.01), or in the week x situation interaction (F(6) = 1.550; *p* = 0.182; ŋp^2^ = 0.16). Table 1 shows the respective results.

The results of the HRmed indicated no difference between the weeks (F(3) = 1.911; *p* = 0.155; ŋp^2^ = 0.19), between the situations (F(2) = 2.051; *p* = 0.161; ŋp^2^ = 0.20), or between the week x situation interaction (F(6) = 1.228; *p* = 0.309; ŋp^2^ = 0.13). The Friedman test also did not show a difference in the HRmax between situations (*p* = 0.478). These results are shown in Table 2 below.

The results of the [BG] detected a difference between the weeks (F(3) = 3.192; *p* = 0.042), between the situations (F(2) = 7.376; *p* = 0.005) in different moments (F(1) = 59.737; *p* = 0.001), and for the situation x moment interaction (F(1) = 15.702; *p* = 0.001), with a large effect size (ŋp^2^ = 0.28; ŋp^2^ = 0.48; ŋp^2^ = 0.88; ŋp^2^ = 0.65, respectively) (Table 3). The BonFerroni posthoc test indicated that these differences occurred between the third and fourth weeks (*p* = 0.021; 95%CI = −8.38. −0.68). The test also indicated that the values of FAST [BG] and FED (*p* = 0.014; 95%CI = −10.73, −1.32) were lower than in the BL (*p* = 0.026; 95%CI = −16.42. −1.10); in addition, regardless of the situation or week, there was a difference in the [BG] values according to the moment analyzed (*p* = 0.001; 95%CI = −46.16. −24.947). Table 2 shows the respective results.

The mean [LAC] reached 3.22 ± 0.89 mM in the pre-test and 16.48 ± 2.17 mM in the post-test for the BL situation. The first week of intervention presented pre- and post-test values of 2.63 ± 0.96 mM and 2.53 ± 0.57 mM and 15.91 ± 3.43 mM and 18.58 ± 2.38 mM, respectively, for the FAST and FED situations. In the fourth week, the pre-test values were 3.05 ± 1.13 mM and 2.57 ± 060 mM, and the post-test values were 15.25 ± 3.82 mM and 18.78 ± 2.94 mM for FAST and FED situations, respectively. Therefore, there was a difference between the pre- and post-test values, with a large effect size (F(1) = 611.577; *p* = 0.001), as well as in the situation x moment interaction (F(2) = 5.076; *p* = 0.020), also with a large effect size (ŋp^2^ = 0.98; ŋp^2^ = 0.38, respectively). The BonFerroni posthoc test indicated differences between the FAST and FED (*p* = 0.047; 95%CI = −2.795. −0.021) scenarios, as well as between the pre- and post-tests (*p* = 0.001; 95%CI = −15.35. −12.73).

### 3.5. Energy Intake

The total calories ingested in the first week were lower than the BL, whereas in the second week, they were greater than in the first week. The total values of calories ingested in the third and fourth weeks were similar but increased compared with the second week. Despite these variations, the total number of calories ingested during IF was lower than the BL. However, no difference between the BL and the intervention weeks was found. In addition, the relative ingested amounts of Kcal/kg, carbohydrate (CHOg/kg), lipid (LIPg/kg), and protein (PROg/kg) also showed no difference (Table 3).

## 4. Discussion

This study aimed to investigate the effect of IF on TBM and performance in Taekwondo. The hypothesis was that the 12 h fasting protocol with breakfast omission would reduce the TBM without decreasing performance. Throughout the four weeks of IF, the results showed a decrease in TBM in the first two weeks, which then remained unchanged in the subsequent weeks. The physical performance was maintained during IF and kept unchanged after the application of the tests, either in a fasted or fed state. Therefore, the study hypothesis was partially proven.

Corroborating these findings, Silva et al. [7] observed that a 12 h acute fast with breakfast omission reduces the TBM of Taekwondo athletes without affecting the CMJ and FSKT tests performance. In our study, the lower TBM in the first two weeks corroborates the efficiency of fasting in a short-term intervention, with limited weight reduction. The athletes might not maintain the IF any longer without a negative energy balance to reduce TBM. Our results show that the mean energy intake decreases initially but increases in the final weeks; however, such a difference is not statistically significant.

Studies with karate athletes performing IF showed a non-significant difference with a small effect size reduction in TBM before (62.4 ± 7.4 kg) and after (61.9 ± 7.1 kg) the Ramadan fast [14]. Corroborating with these results, Zarrouk et al. [10] also found no difference in TBM between pre- (62.1 ± 7.4 kg) and post- (61.8 ± 7.1 kg) evaluation. The authors attribute these results to changes in the diet composition, with an increase in energy and daily water intake. However, herein, we did not observe such a change in the eating behavior of the athletes. Such diverging results might be partially associated with the fasting protocol adopted and Ramadan. In our study, fasting covered the periods of overnight and sleep, when the individuals are less physically active. Therefore, fasting might not have a sufficient impact on energy reserves and hormonal modulation, neither altering the metabolism nor changing eating behavior, as would be expected for IF [9,12,26].

Therefore, the lower TBM during fasting is acute if not associated with a negative energy balance [8,10,14,15]. However, the same sample size with different morphological profiles can have different results. Therefore, a lower TBM with a small effect size in combat sports involving eutrophic athletes might be as important as a large effect size in sedentary obese individuals.

The change in TBM in the first weeks of intervention, without differences either in energy intake or diet composition, may have resulted from the smaller amount of food ingested in the hours before the weight measure, which was performed during the fasting period. Consequently, there is an acute reduction in TBM due to the limited presence of food fragments, nutritional molecules, and water associated with solutes in the gastrointestinal tract during the digestive process [27]. Acute dehydration promoted by fasting might also contribute to the acute reduction in TBM [27]. In addition, the muscle glycogen stores might recover less with a lower water content in the cells [27]. However, such a lower concentration of muscle glycogen may not be sufficient to negatively influence physical performance, considering the low demand for this substrate during the tests performed.

Furthermore, there is an average 28% compensatory response in non-active energy expenditure due to the lower basal energy expenditure, suggesting that only 72% of the extra calories burned in additional activities are extra calories burned that day [28]. Such a compensation in energy expenditure might result from the lower mitochondrial activity [11], which would justify the reduction in TBM in the first weeks, which some athletes then maintained and/or recovered in the last weeks. It is worth noting that our experiment did not cover the measures of glycogen stores, basal energy expenditure, or mitochondrial activity, thus representing a limitation for the interpretation and discussion of the results.

Herein, neither the CMJ nor the FSKT decreased with the application of the IF protocol, either in the FED or FAST states. These results contradict other studies with judokas and karatekas that found significantly lower performance after the tests were applied in the late afternoon during Ramadan [13,15,16]. Those results might be associated with changes in the circadian rhythm due to fasting or the particularity of Ramadan fasting, which may enlarge muscle and liver glycogen stores in the morning rather than in the afternoon [29,30]. Thus, the effect of fasting on physical performance can be influenced by the time of day determined for fasting and the application of the tests. However, other studies applied non-specific tests to Taekwondo and Karate athletes subjected to Ramadan fasting and found no changes in performance, even when conducting the tests in the afternoon [10,14,17]. In contrast, a specific taekwondo test performed at the end of the day indicated a lower TNK during Ramadan fasting [18]. Such diverging results might indicate that not only the time of day can influence the outcomes but also the type of test used, the metabolic demand, and the physical capacity evaluated. Thus, further and more specific studies on this topic should be carried out.

The lower [BG] in the pre-test measure for the FAST state indicates that the volunteers might not have had a meal before the tests. No statistical difference was found in the post-test [BG] between the situations. There was a tendency toward a greater increase in the [BG] for the FAST state compared with the FED. Such a result indicates a potentially greater mobilization of liver glycogen during the test in the FAST state [31]. In addition, the post-test in the FED state also showed higher [LAC] values compared with the FAST, indicating that the lactic anaerobic pathway was demanded and responded in different proportions under different exercise intensities. These data can also indicate less oxidation of substrates for the test applied in the FAST state [31].

Furthermore, the CMJ test, which has a low demand for anaerobic lactic metabolism, might not be sensitive enough to detect performance changes resulting from fasting in Taekwondo athletes [7,17]. Likewise, other studies have demonstrated that either for acute fasting [7] or FED [32], the mean and total of kicks performed in the FSKT did not change. Therefore, even with lower [BG], muscle, and liver glycogen reserves in the pre-test through the fasting protocol, it would also be insufficient to affect performance in a specific modality task. However, the three rounds of the FSKT may have had a greater contribution from aerobic metabolism, as shown in simulated fights [33]. However, this is possibly the only study with a specific test for Taekwondo after a 12 h fast demonstrating a lower performance in the TNK during IF [18]. Contrary to our study, in the aforementioned studies [18,33], the tests were applied in the afternoon. Therefore, there is a lower chance of confirming the hypothesis that a 12 h night IF with the test applied in the morning reduces glycogen stores. Such a scenario can have a lower impact on performance, which seems more plausible than the hypothesis that performance can be affected by the greater aerobic contribution throughout the three rounds of the FSKT test.

Both the HRmed and HRmax for the three FSKT rounds have very high values, as reported in the literature for training sessions, simulated fights, and official competitions [33]. Thus, this indicates that the adaptation in the FSKT test applied in our study reproduced the physiological demand of the modality. In addition, the mean RPE reported after each test round was also maximum or very close to the maximum, indicating that the volunteers performed maximal effort. The KDI did not differ according to the different situations either. No difference in the KDI has also been indicated in studies using the FSKT to rank the level of physical conditioning, competitive level, and/or analyze the effect of different training protocols on performance [22,33]. Therefore, analyzing the KDI may be insufficient to infer either an improvement or worsening in the performance of athletes. Furthermore, all the obtained results should be treated with caution due to the sample size and the variability of ages among the participants in this study.

Further studies should perform the three FSKT rounds using an ergo spirometer to measure the contribution of each energy system and substrate, applied at different times of the day, in both the FAST and FED states. Additionally, it is recommended to incorporate tools and technologies for measuring muscular and hepatic glycogen stores to investigate the effect of intermittent fasting on athletic performance. Furthermore, the effect of intermittent fasting in real and simulated combat situations should be evaluated in future research.

Aware of all the limitations reported throughout the text, the results of the present study can support athletes, coaches, and nutritionists in their practices and prescriptions, according to the competitive calendar and individual objectives. Furthermore, in this study, we aim to advance the scientific discussion regarding intermittent fasting protocols, which are more closely aligned with the cultural reality of Western countries, as well as the evaluation of physical performance measured by non-specific and specific tests. Therefore, it seems that this intermittent fasting protocol can be an efficient strategy for weight loss in athletes over a short period of time, allowing them to fit into the appropriate category without compromising performance.

## 5. Conclusions

Intermittent fasting with 12 h time-restricted feeding and breakfast omission can promote a decrease in total body mass in the first two weeks, with stabilization in the third and fourth weeks. However, this can be promoted without decreasing Taekwondo athletes’ performance.

## Figures and Tables

**Figure 1 nutrients-15-03131-f001:**
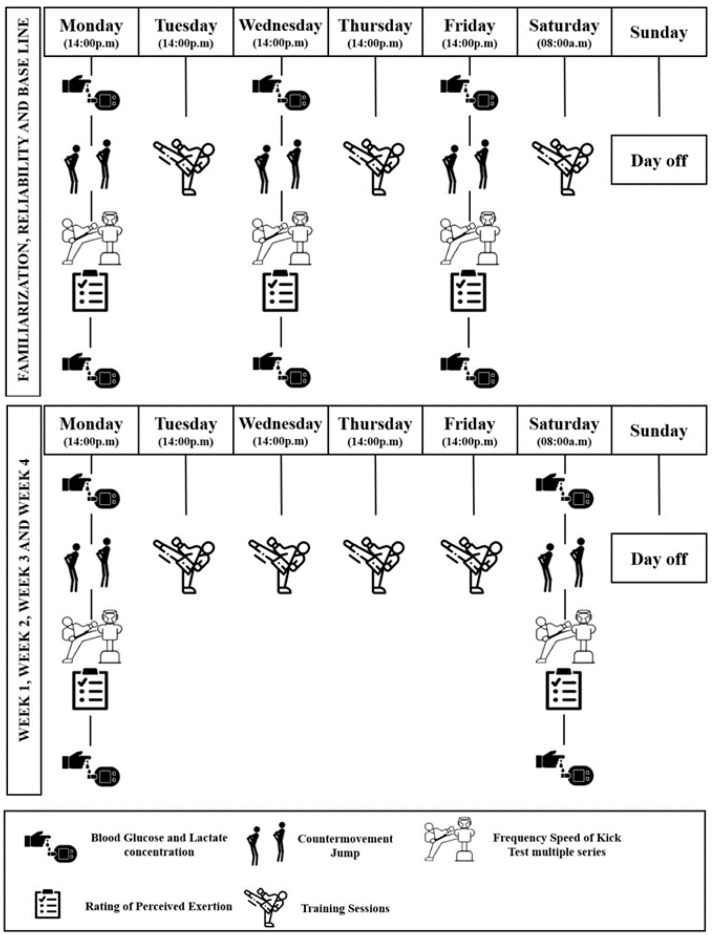
Experimental design.

**Figure 2 nutrients-15-03131-f002:**
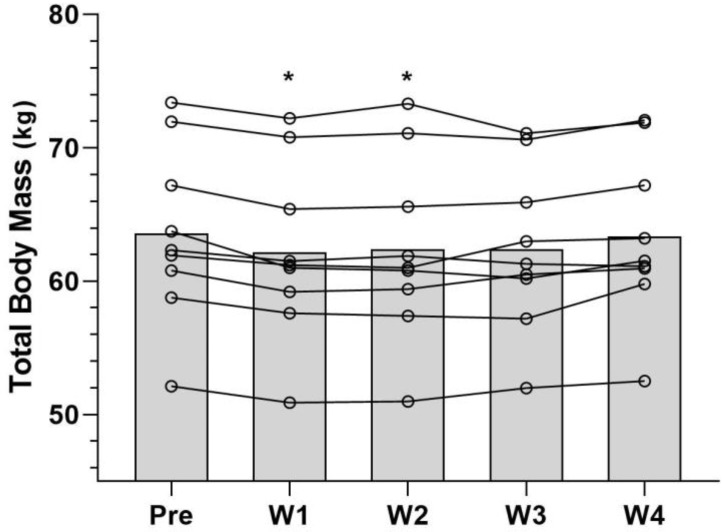
Total body mass variation. Pre: pre-intervention; W1: week 1; W2: week 2; W3: week 3; W4: week 4; * statistically significant difference in relation to the pre-intervention.

**Table 1 nutrients-15-03131-t001:** Performance in CMJ and FSKT.

Variable	Situation	W1	W2	W3	W4
**CMJ_(cm)_**	BL	36.93 ± 6.22			
	FAST	37.36 ± 6.77	36.47 ± 5.78	35.96 ± 5.05	36.91 ± 6.12
	FED	35.26 ± 7.15 *^,&^	37.54 ± 5.79	38.24 ± 6.45 ^&,$^	37.95 ± 5.61 ^$^
**MK**	BL	18.28 ± 0.98			
	FAST	18.17 ± 1.16	18.36 ± 1.17	18.60 ± 0.97	18.80 ± 1.23
	FED	18.23 ± 1.16	18.74 ± 0.84	18.94 ± 1.03	19.04 ± 0.97
**TNK**	BL	274.22 ± 14.82			
	FAST	272.56 ± 15.39	275.44 ± 17.59	279.11 ± 14.66	282.11 ± 18.52
	FED	273. 56 ± 17.40	281.11 ± 12.71	284.11 ± 15.54	285.67 ± 14.58
**RPE**	BL	9.22 ± 0.44			
	FAST	9.33 ± 0.50	9.89 ± 0.33 *	9.89 ± 0.33 *	10.00 ± 0.00 *^,#^
	FED	9.56 ± 0.52	9.56 ± 0.52	9.89 ± 0.33 *	10.00 ± 0.00 *^,#^
**KDI**	BL	8.90 ± 2.91			
	FAST	9.27 ± 3.01	9.27 ± 3.01	9.27 ± 3.01	9.27 ± 3.01
	FED	9.95 ± 2.64	9.95 ± 2.64	9.95 ± 2.64	9.95 ± 2.64

BL: baseline; FAST: fasting state; FED: fed state; MK: mean of kicks of the three rounds of frequency speed of kick test multiple series; TNK: total number of kicks performed in the three rounds of frequency speed of kick tests multiple series. KDI: mean of the fatigue index in the three rounds of frequency speed of kick test multiple series; W1: week 1; W2: week 2; W3: week 3; W4: week 4. * statistically significant difference from baseline; ^&^ statistically significant difference in relation the fasting state; ^$^ statistical difference of week compared to week 1; ^#^ statistically significant difference in comparison to the other weeks.

**Table 2 nutrients-15-03131-t002:** Physiological FSKT variables.

Variable	Situation	W1	W2	W3	W4
**HR mean_(bpm)_**	BL	176.22 ± 7.13			
	FAST	176.00 ± 6.87	171.22 ± 12.86	169.89 ± 8.85	173.56 ± 9.35
	FED	176.33 ± 9.00	171.78 ± 13.67	173.00 ± 8.64	174.11 ± 7.75
**HR max_(bpm)_**	BL	184.59 ± 4.47			
	FAST	185.85 ± 6.73	181.22 ± 11.75	182.89 ± 7.99	183.89 ± 8.10
	FED	185.44 ± 5.87	182.78 ± 7.77	183.78 ± 7.79	185.44 ± 6.34
**[BG]pre_(mg/dL)_**	BL	85.33 ± 9.88 ^&^			
	FAST	70.67 ± 9.06	69.44 ± 10.81	70.22 ± 7.77	69.00 ± 9.68
	FED	81.11 ± 20.43 ^&^	85.67 ± 12.18 ^&^	91.78 ± 14.64 ^&^	97.44 ± 13.93 ^&^
**[BG]post_(mg/dL)_**	BL	120.67 ± 17.22			
	FAST	120.33 ± 13.56 *	115.89 ± 19.21 *	112.22 ± 13.55 *	126.11 ± 17.58 *
	FED	110.56 ± 15.15 *	110.67 ± 18.26 *	108.00 ± 18.20 *	116.89 ± 14.30 *

Note: BL: baseline; FAST: fasting state; FED: fed state; HR mean: Heart Rate mean; HR max: Heart Rate maximal; BGpre: Blood Glucose pre-test; BGpost: Blood Glucose post-test; [LAC]pre: lactate concentration pre-test; [LAC]post: lactate concentration post-test; * statistical difference in relation to the pre-test moment; ^&^ statistical difference in relation to fasting.

**Table 3 nutrients-15-03131-t003:** Caloric intake and nutritional composition of macronutrients.

Variable	Pre	W1	W2	W3	W4	*p* Value	E.S_(_ŋ_p_^2^_)_	Classification
**Kcal**	2211.09 ± 1380.30	1409.41 ± 379.08	1727.69 ± 726.49	1908.93 ± 926.95	1904.00 ± 1462.04	0.121	0.10	Small
**Kcal/kg**	34.71 ± 19.36	22.83 ± 6.99	28.11 ± 11.87	31.38 ± 15.69	30.30 ± 22.11	0.411	0.10	Small
**CHO_(g/kg)_**	4.76 ± 4.29	3.52 ± 1.41	3.86 ± 2.15	3.29 ± 2.14	3.66 ± 2.57	0.540	0.05	Small
**PRO_(g/kg)_**	1.20 ± 0.52	1.28 ± 0.39	1.20 ± 0.37	1.32 ± 0.45	1.32 ± 0.93	0.444	0.01	Small
**LIP_(g/kg)_**	1.04 ± 0.48	0.95 ± 0.38	1.13 ± 0.28	1.08 ± 0.48	0.97 ± 0.54	0.829	0.04	Small

Note: Kcal: total kilocalories; Kcal/kg: kilocalories per kilogram of total body mass; CHO(g/kg): carbohydrate value relative to total body mass, grams per kilogram of body mass; PRO(g/kg): protein value relative to total body mass, grams per kilogram of body mass; LIP(g/kg): lipid value relative to total body mass, grams per kilogram of body mass; Pre: pre-intervention; W1: week 1; W2: week 2; W3: week 3; W4: week 4; E.S (ŋp^2^): square partial ETA effect size.

## Data Availability

All data used within this manuscript are available upon request.

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
