# Peer review of "Intermittent Fasting Promotes Weight Loss without Decreasing Performance in Taekwondo"

_nutrients, 2023, doi:10.3390/nu15143131_

Round 1
Reviewer 1 Report
This small study aimed to investigate the effect of intermittent fasting in the total body mass and the athlete’s performance in Taekwondo. Overall the authors present an interesting study, however a few details are missing. It is also unclear how these finding impact the literature as a whole.
Abstract
Line 17- Please describe why this research question is being explored (i.e., background sentance).
Line 18- Please describe the athletes in greater detail (i.e., age, male/female, etc.)
Line 19- "Is this period" Please correct the typo in this sentance.
Please state that assessments were conducted weekly (if this correct).
Line 33- Please expand upon your conclusions of the impact of fasting on TBM and performance. What impact does this research have on the field.
Materials and Methods
Figure 1- Please add in whan food was consumed. Or add a another figure for data collection proceedures.
Line 112- It states that food intake was recorderd. Please separate this proceedure out as an outcome. Also add additional detail as to how this was collected and subsequently analyzed.
Results
Line 236- HRmed is not described in the methods section, please add how this was colelcted. BG collection and processing is also not decribed. Please add.
Conclusion
Please add additional statements of how this research impacts the research body as a whole. What is the next step? What still needs to be learned?
How does the lack of a control group impact these findings. Why was a control group not involved?
General Comment
There is no need to start a sentance with So,. Please remove these.
Starting sentances with So....please delete this.
Author Response
This small study aimed to investigate the effect of intermittent fasting in the total body mass and the athlete's performance in Taekwondo. Overall the authors present an interesting study. However a few details are missing. It is also unclear how these finding impact the literature as a whole.
We sincerely appreciate your time spent reviewing our article. We value your suggestions and have taken your comments into consideration. We have made the necessary changes to provide more details and clarify the impact of our findings on the literature.
Abstract
Line 17- Please describe why this research question is being explored (i.e., background sentance).
Thank you for your comment, and we understand your observation. We acknowledge that the mentioned sentence in the original text already provides the necessary background to contextualize the research. However, due to the word limitation in the abstract, we were unable to delve further into this aspect.
We appreciate your guidance and recognize the importance of providing appropriate context for the research. We will review the abstract to ensure that all essential information is included within the space constraints.
Line 18- Please describe the athletes in greater detail (i.e., age, male/female, etc.)
We have addressed your request and have included additional details about the athletes in the revised text. We now provide information such as age, gender, and other relevant details to enhance the understanding of the study.
Line 19- "Is this period" Please correct the typo in this sentence. Please state that assessments were conducted weekly (if this correct).
Thank you for your observation. We would like to inform you that we have made the necessary changes to comply with the journal's guidelines, which specify a word limit for the abstract. Furthermore, the mistakes have been corrected.
Line 33- Please expand upon your conclusions of the impact of fasting on TBM and performance. What impact does this research have on the field.
Thank you for your feedback. We apologize for any confusion. We understand the importance of adhering to the guidelines provided by the journal. The abstract is indeed limited to 200 words, and it is recommended to provide an objective representation of the article without exaggerating the main conclusions.
In consideration of these guidelines, we have chosen not to expand upon the conclusions in the abstract. Instead, we have focused on providing a concise overview of the study's aims, methods, and key findings. We believe that the reader can draw more detailed conclusions from the results and discussion presented in the main text.
Materials and Methods
Figure 1- Please add in whan food was consumed. Or add a another figure for data collection procedures
Thank you for your feedback. We would like to inform you that we have addressed your request by inserting the following sentence in the text, before the experimental design figure: "On days of testing in the fed state, the pre-workout meal was consumed between 1 to 2 hours before the start of the tests."
We opted for this strategy because each athlete had their own accustomed meal routine, and adding another scheme to the experimental design figure was making it more difficult to understand. Furthermore, including an additional figure did not necessarily contribute to a better understanding of the experimental design. Therefore, we deemed it more appropriate to include the sentence in the text to provide this important information about the pre-test feeding.
Line 112- It states that food intake was recorder. Please separate this procedure out as an outcome. Also add additional detail as to how this was collected and subsequently analyzed
We have carefully reviewed your comments and made significant changes to the text according to your observations. We have added more information and references to provide a clearer understanding of the food intake recording procedure and its subsequent analysis.
In the revised text we have inserted after the figure 1:
“During the six weeks, the athletes' food intake was recorded weekly over three non-consecutive days [19], followed by a supplementary dietary recall [20], to measure the total calories ingested and the macronutrient composition. Prior to the beginning of the study, the volunteers received explanations and training to record the food intake using homemade measures and photographic records. The supplementary dietary recall was performed based on the weekly food intake record to identify possible errors in the process carried out by the volunteers, such as food and meal identifications, homemade measurements, and timing. Each week, the dietary profile of each volunteer was determined using the supplementary dietary recall, which was adopted to monitor and compare the nutritional intake between the weeks of the experimental intervention. The dietary profile was determined based on the daily total caloric intake and the intake of macronutrients: carbohydrates, lipids, and proteins. All values were normalized by total body mass. These procedures were performed using the DietBox app (Dietbox Informática Ltda, version 7.8.1, Brazil) and the following tables, in order of preference: Brazilian Table of Food Composition (TACO), Nutritional Composition Table of Foods Consumed in Brazil (IBGE), and Sonia Tucunduva Philippi Table.
The results obtained from this process are reported in the Results section, specifically in subsection 3.5 - Energy Intake, along with Table 3, which presents the values for total caloric intake and macronutrient composition.
By incorporating these additional details and references, our aim is to enhance the comprehensiveness of the methodology section and ensure transparency in data collection and analysis.
References inserted:
- Capling, L.; Beck, K.L.; Gifford, J.A.; Slater, G.; Flood, V.M.; O’connor, H. Validity of dietary assessment in athletes: a systematic review. Nutrients. 2017, 9(12), 1-26.
- LO F.P.W; SUN Y.; QIN J.; LO B. Image-based food classification and volume estimation for dietary assessment: a review. IEEE J Biomed Health Inform. 2020, 24(7), 1-14.
Results
Line 236- HRmed is not described in the methods section, please add how this was coleicted.
Thank you for your observation. In the section 2.3 "Familiarization and Reliability" of the manuscript, the sentence you mentioned (line 132) was included to describe the procedures used during the tests. The athletes used a heart rate sensor (Polar H10, Polar Electro Brasil, Ltda) to assess the mean (HRmed) and maximum (HRmax) heart rate in the FSKTmult test, and they also reported their perceived exertion (RPE) immediately after the FSKTmult. These details were included to provide information about the measurements conducted during the study.
BG collection and processing is also not decribed. Please add.
Thank you for your feedback and the opportunity to improve our article. In the 2.2 Experimental Design section, we have included the requested information. The added sentence is as follows: "To avoid performing the tests in a state of hypoglycemia and to verify whether the athletes were in the FED or FAST state, we measured the values of both pre- and post-test blood glucose [BG] concurrently with lactate [LAC] using a FreeStyle Optium Neo glucometer (Abbott Laboratórios do Brasil Ltda, Brazil)".
Conclusion
Please add additional statements of how this research impacts the research body as a whole
We appreciate the reviewer's suggestion. In the last paragraph of the discussion, we have included the following statement: "Aware of all the limitations reported throughout the text, the results of the present study can support athletes, coaches, and nutritionists in their practices and prescriptions, according to the competitive calendar and individual objectives. Furthermore, in this study, we aim to advance the scientific discussion regarding intermittent fasting protocols, which are more closely aligned with the cultural reality of Western countries, as well as the evaluation of physical performance measured by non-specific and specific tests. Therefore, it seems that this intermittent fasting protocol can be an efficient strategy for weight loss in athletes over a short period, allowing them to fit into the appropriate category without compromising performance."
We chose to include these practical application and scientific advancement statements in the discussion to provide a more direct and objective conclusion, aligning with the study's objectives and our interpretation of the findings. We believe that these additions contribute to the broader research body by highlighting the potential benefits of intermittent fasting protocols in the field of sports performance and weight management.
What is the next step? What still needs to be learned?
We would like to inform you that in the revised version of the manuscript, in line 400 of the Discussion section, after some modifications, we included the following sentence: " In future studies, it may be important to perform the three rounds of the FSKT using an ergospirometer to measure the contribution of each energy system and substrate, applied at different times of the day, in FAST or FED. Furthermore, new studies employing tools and technologies for measuring muscular and hepatic glycogen stores are needed to investigate the effect of intermittent fasting on athletic performance. Future studies should also evaluate the effect of intermittent fasting in real and simulated combat situations." This sentence represents our suggestion for the next steps and what still needs to be learned. We believe that this approach can provide valuable insights into the contribution of different energy systems and substrates during the FSKT, considering different time points and nutritional states.
How does the lack of a control group impact these findings? Why was a control group not involved?
We would like to add that we understand the limitation of not having a control group in the study. However, given that our participants are high-level athletes in the pre-competition period, it was challenging to find a sample that was not undergoing any form of intervention to reduce total body mass. This could have introduced confounding factors in the analysis of the data regarding the proposed objective. Therefore, we chose to conduct a paired-design experiment where each individual serves as their own control.
Comments on the quality of English language. Starting sentences with So…. Please delete this.
Thank you for your comments and suggestions regarding our manuscript. We have carefully reviewed your feedback and made the necessary revisions. We have replaced all instances of the word "So" in the text with appropriate alternatives such as "Therefore," "Thus," "Consequently," "Accordingly," "Hence," "As a result," or "In this regard." We believe these modifications have improved the clarity and flow of the manuscript. We appreciate your valuable input and have taken it into account to enhance the quality of our work.
Additionally, we have submitted our article for English language revision by a professional expert in translations, transcriptions, and revisions. The certificate will be attached for verification.

Reviewer 2 Report
I have attached manuscript with edits

This paper cannot be assessed in its current form. I encourage you to work with a colleague whose first language is English. This is critical! Moreover, please document clearly in the introduction how this paper differs from the literature, i.e., what is new here?
Author Response
Dear reviewer,
We sincerely appreciate your time and valuable feedback on our article. Based on your suggestions, we have made the necessary corrections, including improvements in the spelling and grammar. Additionally, we have submitted our article for English language revision by a professional expert in translations, transcriptions, and revisions. The certificate will be attached for verification.
Line 1. Title.
The title has been changed to: "Intermittent Fasting Promotes Weight Loss without Decreasing Performance in Taekwondo."
Thanks.
Line 40. Introduction.
Based on your suggestion, we have made the requested modification in the sentence. Therefore, we have removed "what it means without restrictions." The new sentence now reads as follows: " Intermittent fasting (IF) is the partial or total food intake abstinence in a predetermined period of 12 to 48 hours with little or no caloric intake interspersed with non-restricted feeding"
Line 45-47. Introduction.
The sentence has now been revised to: “However, fasting exerts some effects that might lower sports performance, such as sleep disorders, hypohydration, low motivation, and reduced muscle glycogen stores [9-10]. The caloric restriction that is common in fasting can lower the mitochondrial intrinsic activity without altering the mitochondrial content in skeletal human muscle [11]. Moreover, longer IF periods can increase cortisol levels and stimulate hunger, which, in turn, can cause the TBM to be either maintained or increased [12]."
This sentence is now located in lines 50-55.
We once again thank you for your feedback and contribution to improving the quality of our work.
Line 49.
Thank you for your feedback. We have made the necessary correction in the sentence you mentioned. It now reads as follows: "Moreover, longer IF periods can increase cortisol levels and stimulate hunger, which, in turn, can cause the TBM to be either maintained or increased".
This sentence is now located in lines 53-55.
Line 55-57
We have reviewed your suggestion and made the necessary correction in the sentence. It now reads as follows: "However, in others, Ramadan fasting decreased performance in combat sports."
This sentence is now located in lines 59-60.
Line 60-61
We have taken into account your suggestion and made the necessary change throughout the entire text where the word "taekwondo" appeared with a lowercase "t." We have now corrected all instances to "Taekwondo" with an uppercase "T," in accordance with the appropriate nomenclature for this martial art.
Line 68-70.
Thank you for your time and valuable feedback on our article. We have taken your suggestion into consideration and made the necessary change. Specifically, we have replaced "IN" with "ON" as requested. The revised sentence now reads as follows: "Therefore, it is necessary to investigate other intermittent fasting (IF) chronic protocols on physical performance." We appreciate your attention to detail and your contribution to improving the clarity of our work.
- Materials and Methods
Might be better to study separately as men and women may differ in their responses.
Thank you for your comments and suggestions regarding our study. We would like to inform you that we have made the necessary updates to the text as suggested by the reviewers. In the discussion section, we have included the limitations of the study, including the sample size and gender differences. Additionally, we understand that men and women may respond differently.
Therefore, the limitations mentioned by the reviewers have been duly reported in the text, aiming for greater transparency and understanding of the results.
We would like to inform you that we conducted separate analyses for men and women to investigate potential differences in their responses. However, the results obtained did not show significant differences compared to the analysis presented in the study. We acknowledge that the sample size is a limitation of our study, which may affect the detection of subtle differences between groups. However, due to the matched design of the study, we believe that the potential issue of men and women responding differently has been minimized.
Line 78-80. Methods.
As per your suggestion, we have replaced the word "weight" with "masses." The revised sentence now reads as follows: " Our inclusion criteria covered those who were competing and kept their body masses stable at least two months before the research period" We appreciate your attention to detail and your contribution to improving the accuracy of our work.
This sentence is now located in lines 85-87
Line 132-133
As per your suggestion, we have replaced the word "performed" with "perform" for improved clarity. The revised sentence now reads as follows: "The test was conducted using Boomboxe® (Simulacare, São Paulo – Brazil) and the athletes were instructed to perform the maximum number of kicks possible."
This sentence is now located in lines 151-152
Line 169. Statistical analysis.
As per your suggestion, we have replaced "will be" with "was" for improved accuracy. The revised sentence now reads as follows: "All data are presented as mean, standard deviation, and 95% confidence interval (CI 95%)." We appreciate your attention to detail and your contribution to enhancing the quality of our work.
This sentence is now located in lines 191-192
Results
Figure 2. Best use a scale that shows significant results as different
Thank you for your comment regarding Figure 2 in our study. We would like to inform you that we have made the necessary adjustment to the scale used in the figure to highlight significant results as different. We have taken into consideration the main objective of the graph, which is to illustrate individual differences. The revised scale now clearly and distinctly represents the relevant findings, allowing for better visualization of variations among individuals. We appreciate your observation and contribution to improving the presentation of data in our study.

Reviewer 3 Report
The subject and purpose of the research are very important in the context of the widespread reduction of body weight in combat sports. The authors emphasize the ambiguity of published research results by other authors. Therefore, any attempt to scientifically verify the common weight regulation practice deserves attention and dissemination. In the introduction, you can add 2-3 sentences about rapid weight loss and side effects - a common practice in combat sports. I suggest modifying the statement: "This is a common nutritional strategy used by Taekwondo athletes for weight loss". The obtained results should be treated with caution (it should be emphasized), because only 2 women and 7 men of significantly different ages (18.4 ± 3.3 years), i.e. also juniors, were studied. The effects of the experiment were assessed by tests and did not take into account behavior in combat (even simulated) - this should also be emphasized. The material deserves to be published!
Author Response
The subject and purpose of the research are very important in the context of the widespread. Suggestions for Authors reduction of body weight in combat sports. The authors emphasize the ambiguity of published research results by other authors. Therefore, any attempt to scientifically vinify the common weight regulation practice deserves attention and dissemination.
Thank you for your comment regarding the importance of the subject and the objective of our research in the context of widespread body weight reduction in combat sports. We fully agree with emphasizing the ambiguity of research results published by other authors. Therefore, we believe that any attempt to scientifically verify the common weight regulation practice deserves attention and dissemination.
We have taken note of your considerations and have revised the text to further highlight the significance of this topic and the need to promote the dissemination of the findings. We hope that these changes meet your expectations and contribute to the acceptance of our article in this journal.
In the introduction. you can add 2-3 sentences about rapid weight loss and side effects - a common practice in combat sports.
Thank you for your suggestions. Taking your comment into consideration, we have included 2-3 sentences about rapid weight loss and its effects in the first paragraph of the introduction. The text now states that in combat sports, there are numerous strategies for rapid reduction of total body mass, such as hypocaloric diets, water restriction, use of laxatives, immersion in hot water baths, sauna sessions, increased quantity and duration of training sessions, and performing exercises with heavy clothing and/or plastic suits [1]. However, we emphasize that these strategies can have detrimental effects on athletic performance, including muscle mass reduction, decreased overall and specific endurance, impaired training adaptations, decreased concentration capacity, difficulty in decision-making, mood alterations such as increased irritability and demotivation, and increased discomfort and subjective perception of effort following training sessions [2].
We hope that these modifications meet your requests and contribute to the improvement of the article.
References inserted:
- Andreato, L.V.; Andreato, T.V.; Santos, J.F.S.; Esteves, J.V.D.C.; Moares, S.M.F.; Franchini, E. Weight loss in mixed martial arts athletes. J Combat Sports Martial Arts. 2014, 5(2), 125-131.
- Artioli, G. G., Saunders, B., Iglesias, R. T., & Franchini, E. It is time to ban rapid weight loss from combat sports. Sports Med, 2016, 46(11), 1579-1584.
I suggest modifying the statement: This is a common nutritional strategy used by Taekwondo athletes for weight loss. The obtained results should be treated with caution it should be emphasized). because only 2 women and 7 men of significantly different ages (18.4 ‡ 3.3 years], i.e. also juniors were studied. The effects of the experiment were assessed by tests and did not take into account behavior in real or simulated combat.
Thank you for your suggestions and comments on our article. We would like to clarify that the statement "Thereby, it is a common for Taekwondo athletes to use it as nutritional strategy for weight loss" is based on the referenced articles:
- Janiszewska, K.; Przybylowicz, K. Pre-competition weight loss among Polish Taekwondo competitors – occurrence, methods and health consequences. Arch Budo. 2015, 11, 41–45.
- Santos, J.F.S.; Takito, M.Y.; Artioli, G.G.; Franchini, E. Weight loss practices in Taekwondo athletes of different competitive levels. J Exerc Rehab. 2016, 12, 202–208.
These references support the information that the mentioned nutritional strategy is commonly used by Taekwondo athletes for weight loss. However, we understand the importance of emphasizing the need for caution in interpreting the results, as the study involved only 2 women and 7 men with significantly different ages (18.4 ± 3.3 years). This information has been added in the discussion section, in a statement addressing the study's limitations.
Furthermore, we have inserted a paragraph at the end of the discussion with suggestions for future research that considers the evaluation of the effects of intermittent fasting in real and simulated combat situations.
We have taken note of these considerations and made the necessary changes to make the statement more accurate and well-founded. We appreciate your contribution and hope that the modifications meet your expectations for the acceptance of our article in this journal.
Round 2
Reviewer 1 Report
No additional edits/comments.
Reviewer 2 Report
There still a few problems with the English but over all the paper is much better. Further, the authors have addressed my other concerns.
Still a few concerns.